# Lipid hydroperoxides promote sarcopenia through carbonyl stress

Hiroaki Eshima[1,2,3†], Justin L Shahtout[1,4†], Piyarat Siripoksup[1,4], MacKenzie J Pearson[5], Ziad S Mahmassani[1,2,4], Patrick J Ferrara[1,2,6], Alexis W Lyons[1], John Alan Maschek[1,6,7], Alek D Peterlin[1,6], Anthony RP Verkerke[1,6], Jordan M Johnson[1,6], Anahy Salcedo[1], Jonathan J Petrocelli[1,4], Edwin R Miranda[1,2], Ethan J Anderson[8], Sihem Boudina[1,2,6], Qitao Ran[9], James E Cox[1,7,10], Micah J Drummond[1,2,4], Katsuhiko Funai[1,2,4,6]*

[1]Diabetes and Metabolism Research Center, University of Utah, Salt Lake City, United States; [2]Molecular Medicine Program, University of Utah, Salt Lake City, United States; [3]Department of International Tourism, Nagasaki International University, Nagasaki, Japan; [4]Department of Physical Therapy & Athletic Training, University of Utah, Salt Lake City, United States; [5]Sciex, Framingham, United States; [6]Department of Nutrition & Integrative Physiology, University of Utah, Salt Lake City, United States; [7]Metabolomics Core Research Facility, University of Utah, Salt Lake City, United States; [8]Fraternal Order of Eagles Diabetes Research Center, University of Iowa, Iowa City, United States; [9]Department of Cell Systems and Anatomy, The University of Texas Health Science Center at San Antonio, San Antonio, United States; [10]Department of Biochemistry, University of Utah, Salt Lake City, United States

*For correspondence:
kfunai@utah.edu

†These authors contributed equally to this work

**Abstract** Reactive oxygen species (ROS) accumulation is a cardinal feature of skeletal muscle atrophy. ROS refers to a collection of radical molecules whose cellular signals are vast, and it is unclear which downstream consequences of ROS are responsible for the loss of muscle mass and strength. Here, we show that lipid hydroperoxides (LOOH) are increased with age and disuse, and the accumulation of LOOH by deletion of glutathione peroxidase 4 (GPx4) is sufficient to augment muscle atrophy. LOOH promoted atrophy in a lysosomal-dependent, proteasomal-independent manner. In young and old mice, genetic and pharmacological neutralization of LOOH or their secondary reactive lipid aldehydes robustly prevented muscle atrophy and weakness, indicating that LOOH-derived carbonyl stress mediates age- and disuse-induced muscle dysfunction. Our findings provide novel insights for the role of LOOH in sarcopenia including a therapeutic implication by pharmacological suppression.

## Editor's evaluation

This paper is of fundamental importance for its description of the role of lipid peroxidation in the loss of muscle mass and contractile function during aging and hindlimb suspension. The evidence is generally solid though is incomplete in some areas. The paper will be of particular interest to those who study the biology of aging-related muscle dysfunction.

## Introduction

Loss of muscle mass and function with age is detrimental to health and quality of life (*Evans, 2010*; *Larsson et al., 2019*). Sarcopenia, muscle atrophy and weakness with aging, is due to a combination of inactivity, injury, surgery, and biological consequences of aging (*Bonaldo and Sandri, 2013*;

*Dolbow and Gorgey, 2016*). A pharmacological therapy for muscle loss does not exist, and current diet or exercise therapeutic approaches are often ineffective or unfeasible. Oxidative stress has been implicated in muscle atrophy by accelerating proteolysis (*Powers et al., 2011*; *Scicchitano et al., 2018*), but the exact mechanism by which reactive oxygen species (ROS) contributes to the decrease in muscle mass and strength is not well understood.

Lipid hydroperoxide (LOOH) is a class of ROS molecules that has been implicated in cell damage, particularly as a trigger to induce ferroptosis, a non-apoptotic form of regulated cell death (*Wiernicki et al., 2020*; *Yang et al., 2014*). Lipid peroxidation is initiated by prooxidants such as hydroxyl radicals attacking the carbon-carbon double bond in fatty acids, particularly the polyunsaturated fatty acids (PUFAs) containing phospholipids (*Bochkov et al., 2010*). Lipid radicals (L•) created by this reaction rapidly react with oxygen to form a lipid peroxy-radical which subsequently reacts with another lipid to produce L• and LOOH, the former propagating lipid peroxidation. LOOH is the primary product of lipid peroxidation that forms secondary reactive lipid aldehydes such as 4-hydroxynonenal (4-HNE) and malondialdehyde (MDA), inducing carbonyl stress with high reactivity against biological molecules to promote cellular toxicity. The intracellular level of LOOH is endogenously suppressed by glutathione peroxidase 4 (GPx4) that catalyzes the reaction by which LOOH is reduced to its nonreactive hydroxyl metabolite (*Anderson et al., 2018*).

Despite the evidence for the role of LOOH-mediated cell damage and cell death, the biological consequence of LOOH accumulation in skeletal muscle is not well understood (*Bhattacharya et al., 2009*; *Pharaoh et al., 2020*). Below, we provide evidence that LOOH mediates loss of muscle mass and function associated with sarcopenia. An increase in muscle LOOH was a common feature with aging and disuse, and accumulation of LOOH in vitro and in vivo augmented muscle atrophy. We further show that genetic or pharmacological suppression of LOOH and their reactive lipid aldehydes is sufficient to prevent disuse-induced muscle atrophy, and to a greater extent in muscle contractile function, in young and old mice.

## Results

We first evaluated the changes in skeletal muscle LOOH with aging. In humans and in mice, aging promoted a reduction in the expression of *Gpx4* in skeletal muscle (*Figure 1A, B*). To examine the changes in skeletal muscle LOOH landscape with age, we performed a comprehensive oxidolipidomic analysis in gastrocnemius muscle samples from young (4 months) and old (20 months) mice (*Figure 1C, D*). We detected over 300 species of oxidized lipids with an effect distribution that was highly class-dependent. Among these, age had the most robust effect on oxidized phosphatidylethanolamine (*Figure 1C*, red), a class of lipids that have been implicated as a potential lipid signal to induce ferroptosis (*Kagan et al., 2017*). Among the top ten oxidized lipid species whose abundance had the most robust fold-increase with age, six of them were oxidized phosphatidylethanolamine species (*Figure 1D*). These oxidized phosphatidylethanolamine species were also substantially more highly abundant compared to other oxidized lipids. LOOH can be indirectly assessed by quantifying lipid aldehyde adducts such as 4-HNE and MDA. We confirmed increased muscle 4-HNE and MDA with age (*Figure 1E–G*).

Disuse promoted by inactivity, injury, or surgery is a major contributor to age-associated decline in muscle mass and function. Disuse also promotes skeletal muscle atrophy that is likely contributed by ROS (*Powers et al., 2011*). To model disuse atrophy, mice underwent a hindlimb unloading (HU) procedure as previously described (*Eshima et al., 2020*; *Heden et al., 2019*; *Figure 1—figure supplement 1A–J*). As expected, HU-induced muscle atrophy and weakness (*Figure 1—figure supplement 1D, E*) concomitant to reduction in body and lean mass (*Figure 1—figure supplement 1B, C*). Disuse robustly elevated muscle LOOH levels (*Figure 1—figure supplement 1F, G*) without significant changes in mitochondrial ROS production (*Figure 1—figure supplement 1H-J*). An increase in muscle LOOH preceded atrophy (*Figure 1—figure supplement 1D, F*), consistent with the notion that LOOH may trigger mechanisms to promote loss of muscle mass.

Next, we tested our hypothesis that LOOH contributes to muscle atrophy using C2C12 myotubes (*Figure 2A*). Lentivirus-mediated knockdown (KD) of GPx4 increased LOOH and markers of ferroptosis concomitant with a decrease in myotube diameter (*Figure 2B–I*). We also recapitulated these findings with erastin (a system $X_c^-$ inhibitor that suppresses glutathione synthesis) (*Figure 2B, C* and *Figure 2—figure supplement 1A–E*) and RSL3 (GPx4 inhibitor) (*Figure 2B, C* and *Figure 2—figure supplement*

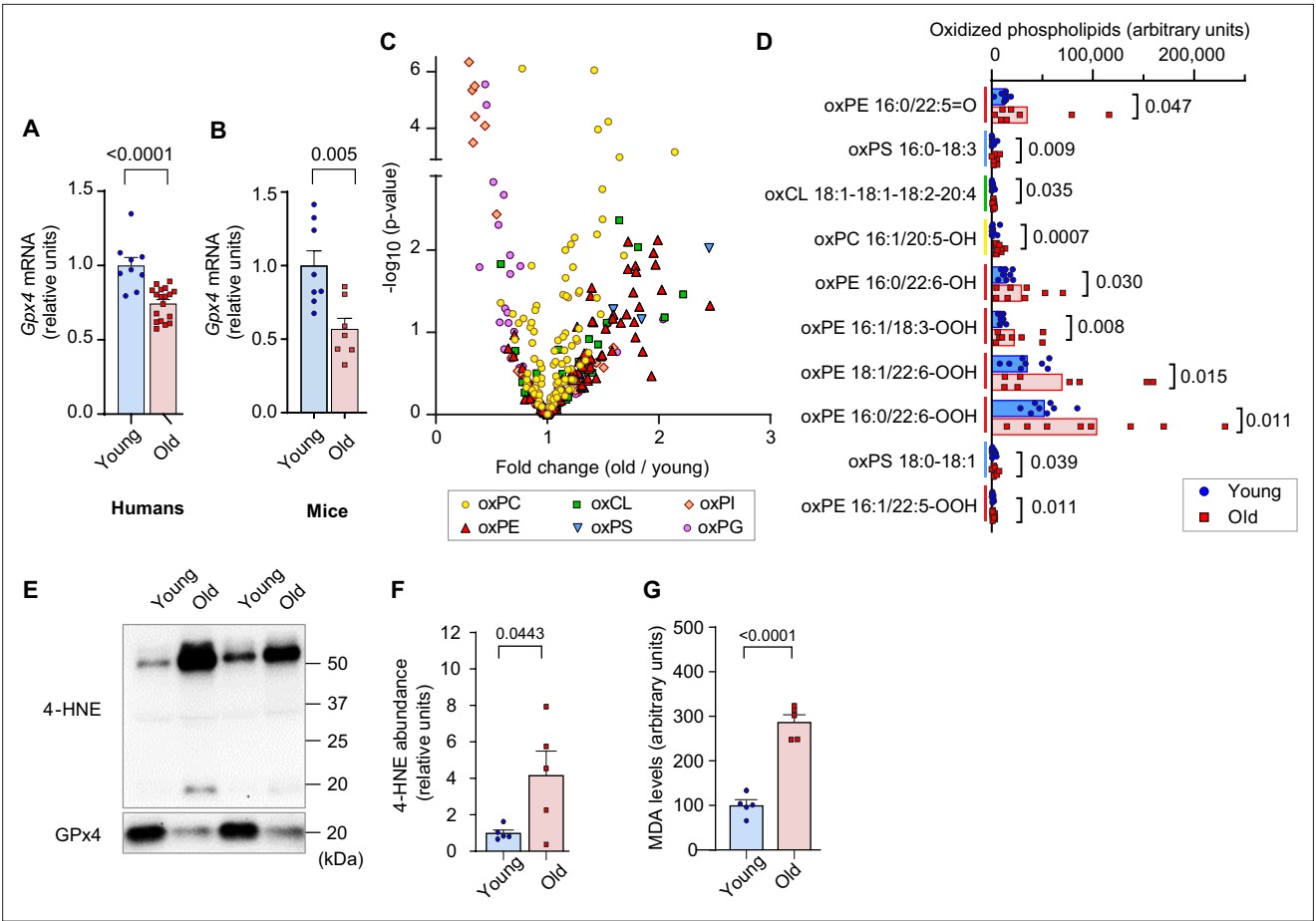

**Figure 1.** Lipid hydroperoxides (LOOH) increases with age in skeletal muscle. (**A, B**) Glutathione peroxidase 4 (*Gpx4*) mRNA levels in skeletal muscle biopsy samples from young and old humans (n=9 young [2 men and 7 women], n=18 for old [11 men and 7 women]) (**A**) or gastrocnemius muscles from young and old mice [n=8 for young, n=7 for old] (**B**). (**C, D**) Oxidized phospholipid content in gastrocnemius muscles from young and old male mice (n=8 per group). (**E, F, G**) Immunoblotting (**E**) and quantification (**F**) of 4-hydroxynonenal (4-HNE) proteins (n=5 per group) and malondialdehyde (MDA) levels (**G**) (n=3 per group) in gastrocnemius muscles. Male mice were used. Data are shown as the mean ± SEM. Statistical analyses in (**A, B, C, D, F**) and (**G**) were performed with an unpaired two-tailed t-test.

The online version of this article includes the following source data and figure supplement(s) for figure 1:

**Source data 1.** Original western blots.

**Figure supplement 1.** Disuse promotes lipid hydroperoxides (LOOH) in skeletal muscle.

**Figure supplement 1—source data 1.** Original western blots.

*1F–I*), commonly used acute pharmacological interventions to elevate intracellular LOOH. These data support the idea that LOOH reduces myotube size in a cell-autonomous manner.

We then translated these findings in vivo with global heterozygous GPx4 knockout mice (*Gpx4*<sup>+/-</sup>). Germline deletion of GPx4 is embryonically lethal (*Yant et al., 2003*), but *Gpx4*<sup>+/-</sup> mice appear normal and do not have an observable muscle phenotype at baseline (*Anderson et al., 2018*; *Katunga et al., 2015*). We studied 4 months (young) and 20 months (old) *Gpx4*<sup>+/-</sup> and wildtype littermates with or without HU (*Figure 3—figure supplement 1A–D*). In young mice, GPx4 haploinsufficiency augmented the loss in soleus mass induced by HU (*Figure 3A* and *Figure 3—figure supplement 1E*). However, muscle masses between old *Gpx4*<sup>+/-</sup> and wildtype mice were not different. We interpret these findings to mean that disuse in old mice promotes an increase in LOOH that has already reached a maximally effective threshold with age such that GPx4 deletion had no further effect. In support of this, we saw no differences in 4-HNE or MDA levels between old *Gpx4*<sup>+/-</sup> and wildtype mice (*Figure 3B* and *Figure 3—figure supplement 1F, G*). GPx4 haploinsufficiency did not alter force-generating capacity (*Figure 3C* and *Figure 3—figure supplement 1H–J*).

Cell Biology

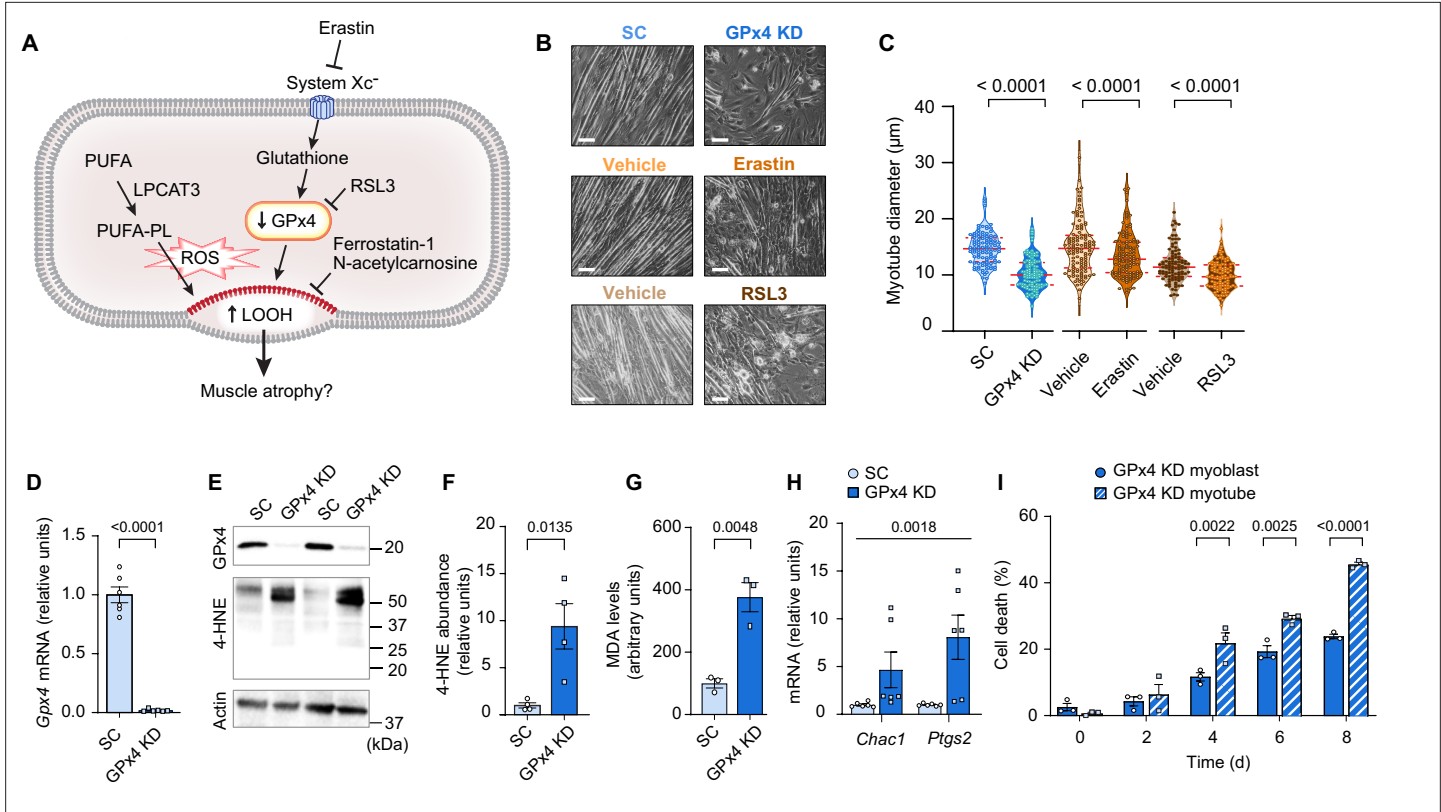

**Figure 2.** Elevated lipid hydroperoxides (LOOH) is sufficient to promote atrophy in cultured myotubes. (**A**) A schematic of how pathways that regulate LOOH may promote muscle atrophy. PUFA: phospholipids containing polyunsaturated fatty acids. (**B, C**) Representative images (**B**) and quantification (**C**) of myotube diameter (n=104 for scrambled: SC, n=107 for glutathione peroxidase 4 [GPx4] knockdown [KD], n=117 for vehicle, n=120 for erastin, n=104 for vehicle, n=110 for RSL3). Scale bar, 100 µm. (**D**) *Gpx4* mRNA levels in C2C12 myotubes with or without GPx4 knockdown (GPx4 KD) (n=6 per group). (**E, F, G**) Immunoblotting of 4-hydroxynonenal (4-HNE), GPx4, and actin (**E**), quantification of 4-HNE (**F**) proteins and malondialdehyde (MDA) levels (**G**) (n=3 per group). (**H**) mRNA levels for *Chac1* and *Ptgs2*, markers of ferroptosis (n=6 per group). (**I**) Cell death levels in GPx4 KD myoblast or myotubes (n=3 independent repeats). Data are shown as the mean ± SEM. Statistical analyses in (**C, D, F, G**) and (**I**) were performed with an unpaired two-tailed t-test. Statistical analyses in (**H**) were performed with a two-way analysis of variance (ANOVA) and Tukey's multiple comparison test.

The online version of this article includes the following source data and figure supplement(s) for figure 2:

**Source data 1.** Original western blots.

**Figure supplement 1.** Elevated lipid hydroperoxides (LOOH) is sufficient to promote atrophy in cultured myotubes.

**Figure supplement 1—source data 1.** Original western blots.

Because GPx4 is expressed globally, we also studied mice with skeletal muscle-specific tamoxifen-inducible GPx4 knockout (GPx4-MKO) (*Figure 3D* and *Figure 3—figure supplement 2A*; *Yoo et al., 2012*). Consistent with *Gpx4*[+/-] mice, soleus muscles from GPx4-MKO mice were also more prone to developing disuse-induced atrophy (*Figure 3E and F* and *Figure 3—figure supplement 2B–F*) concomitant to elevated LOOH (*Figure 3G* and *Figure 3—figure supplement 2G, H*), suggesting that loss of GPx4 in muscle augments atrophy in a cell-autonomous manner. Histological analyses revealed that reduced muscle mass was consistent with reduced cross-sectional area (CSA) of myofibers regardless of fiber-type compositions (*Figure 3H&I* and *Figure 3—figure supplement 2I, J*). These data implicate that LOOH directly reduces muscle cell size in vivo.

GPx4 primarily neutralizes LOOH but it also exhibits some activity toward other peroxides (*Imai and Nakagawa, 2003*). To confirm that the effects of GPx4 deletion to promote atrophy is specific to LOOH, we diminished the ability of cells to incorporate PUFAs into phospholipids by deleting lysophosphatidylcholine acyltransferase 3 (LPCAT3) (*Lee et al., 2021*; *Ferrara et al., 2021a*; *Ferrara et al., 2021b*). LPCAT3 is an enzyme of Lands cycle that preferentially acylates lysophospholipids with PUFAs, and thus an essential component of ferroptosis (*Kagan et al., 2017*). Indeed, LPCAT3 KD rescued the increase in 4-HNE induced by GPx4 KD (*Figure 4A–C* and *Figure 4—figure supplement*

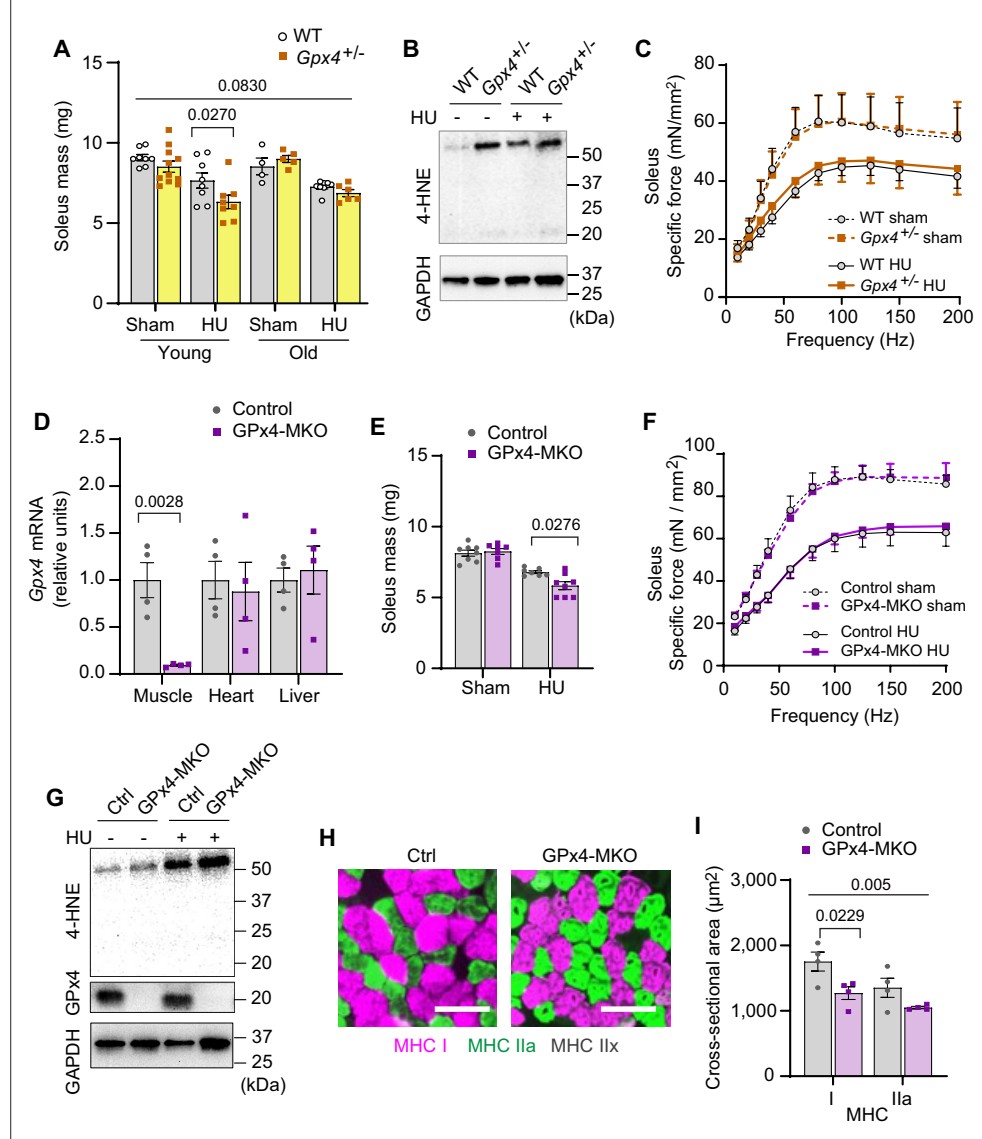

**Figure 3.** Elevated lipid hydroperoxides (LOOH) is sufficient to augment disuse-induced muscle atrophy in young and old mice. (**A**) Soleus muscle mass from young or old wildtype (WT) or *Gpx4*$^{+/-}$ mice with or without hindlimb unloading (HU) (n=8–11 per young group, n=4–8 per old mice group). (**B**) Immunoblotting of 4-hydroxynonenal (4-HNE) from gastrocnemius muscles in old WT or *Gpx4*$^{+/-}$ mice. (**C**) Force-frequency curve from old WT or *Gpx4*$^{+/-}$ mice (n=4–7 per group). (**D**) mRNA levels of *Gpx4* from young control or GPx4-MKO mice (n=4 per group). (**E, F**) Soleus muscle mass (**E**) (n=7–8 per group) or force-frequency curve (**F**) from young control or GPx4-MKO mice (n=4–7 per group). (**G**) Immunoblotting of 4-HNE and GPx4 from gastrocnemius muscles in young GPx4-MKO. (**H, I**) Representative images of MHC immunofluorescence (**H**) and muscle fiber cross-sectional area (CSA) by fiber type (**I**) for soleus muscles in young control or GPx4-MKO mice with HU (n=4 per group). Scale bar, 100 µm. Data from *Gpx4*$^{+/-}$ and GPx4-MKO experiments are from male and female mice. Data are shown as the mean ± SEM. Statistical analyses in (**D**) were performed with an unpaired two-tailed t-test. Statistical analyses in (**A, C, E, F**) and (**I**) were performed with a two-way analysis of variance (ANOVA) and multiple comparisons were performed using Tukey's (**C, E, F, I**) or Sidak's (**A**) multiple comparisons tests. GPx4, glutathione peroxidase 4.

The online version of this article includes the following source data and figure supplement(s) for figure 3:

**Source data 1.** Original western blots.

**Figure supplement 1.** Additional data from *Gpx4*$^{+/-}$ mice.

**Figure supplement 2.** Additional data from GPx4-MKO mice.

**Figure supplement 2—source data 1.** Original genotyping gels.

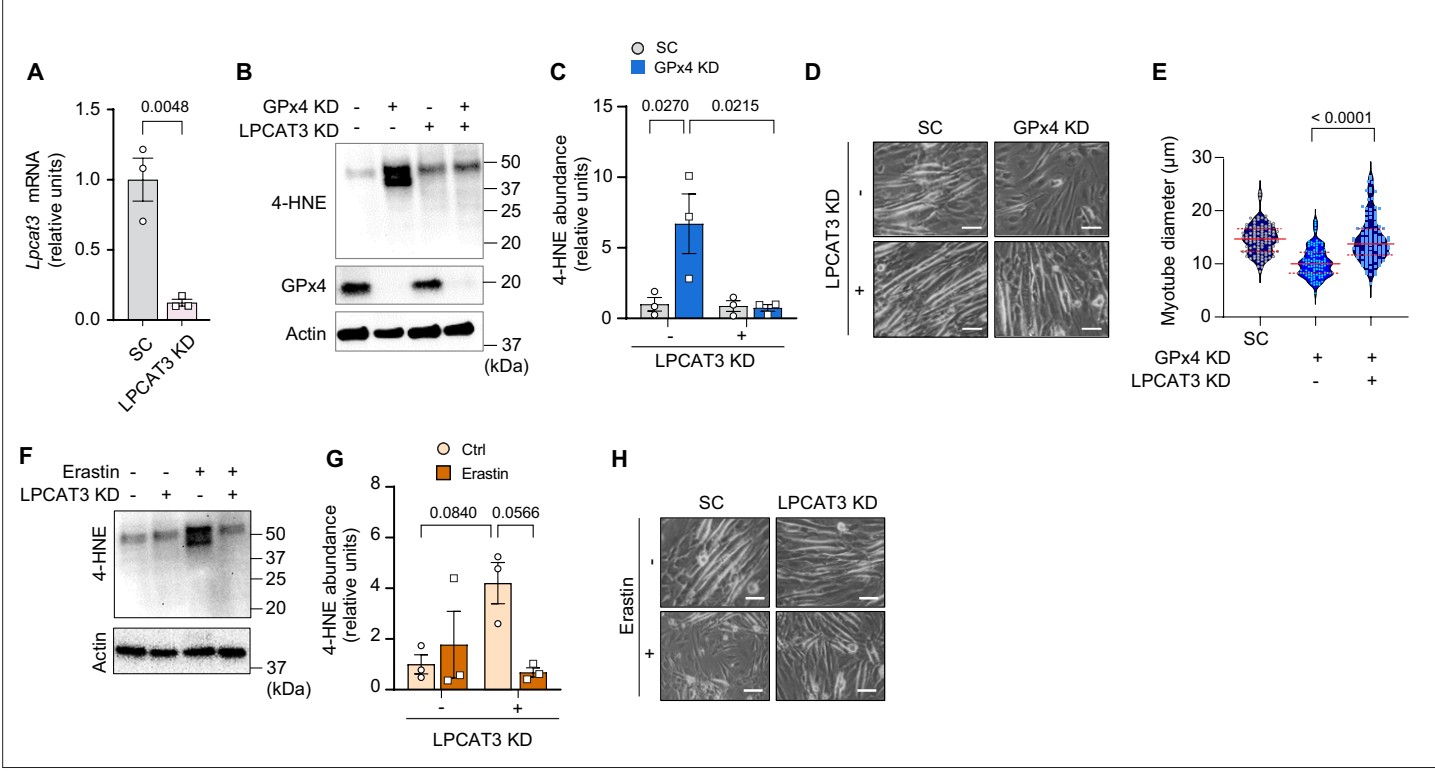

**Figure 4.** Suppression of polyunsaturated fatty acid (PUFA) incorporation prevents lipid hydroperoxides (LOOH)-induced myotube atrophy. (**A**) mRNA levels of lysophosphatidylcholine acyltransferase 3 (*Lpcat3*) in C2C12 myotubes with or without LPCAT3 knockdown (KD) (n=3 per group). (**B**) Immunoblotting of 4-hydroxynonenal (4-HNE), glutathione peroxidase 4 (GPx4), and actin protein in C2C12 myotubes with or without GPx4 KD and/or LPCAT3 KD. (**C**) Quantification of 4-HNE proteins in C2C12 myotubes with or without LPCAT3 KD and/or GPx4 KD (n=3 per group). (**D, E**) Representative images (**D**), and quantification of myotube diameter (**E**) from C2C12 myotubes with or without GPx4 KD and/or without LPCAT3 KD (n=104–114 per group). Scale bar, 100 µm. (**F, G**) Immunoblotting (**F**) and quantification (**G**) of 4-HNE from C2C12 myotubes with or without LPCAT 3 KD and/or erastin (n=3 per group). (**H**) Representative images from C2C12 myotubes with or without LPCAT3 KD and/or erastin (n=3 independent repeats). Scale bar, 100 µm. Data are shown as the mean ± SEM. Statistical analyses in (**A**) were performed with an unpaired two-tailed t-test. Statistical analysis in (**C**) and (**G**) were performed with a two-way analysis of variance (ANOVA) and multiple comparisons were performed using Tukey's multiple comparisons tests. Statistical analyses in (**E**) was performed with a one-way ANOVA with Dunnett's multiple comparisons test.

The online version of this article includes the following source data and figure supplement(s) for figure 4:

**Source data 1.** Original western blots.

**Figure supplement 1.** Additional data from lysophosphatidylcholine acyltransferase 3 (LPCAT3) knockdown (KD).

*1A*). Remarkably, deletion of LPCAT3 KD completely restored the reduction in myotube diameter induced by GPx4 KD (*Figure 4D&E*). Similarly, LPCAT3 deletion also prevented LOOH and cell death induced by erastin (*Figure 4F–H* and *Figure 4—figure supplement 1B, C*). These findings indicate that muscle atrophy induced by loss of GPx4 or erastin treatment is due to the accumulation of LOOH and not other peroxides.

What is the mechanism by which LOOH promotes muscle atrophy? C2C12 myotubes were pretreated with bafilomycin A1 (BafA1) or MG132 prior to erastin incubation to determine whether LOOH increases protein degradation in a lysosomal- or proteasomal-dependent manner, respectively. Erastin-induced reduction in myotube diameter was suppressed with BafA1, but not with MG132 (*Figure 5A, B* and *Figure 5—figure supplement 1A*), suggesting that the lysosome mediates protein degradation by LOOH (*Gao et al., 2018*). We also reproduced these findings with RSL3 treatment (*Figure 5—figure supplement 1B, C*). How does LOOH, a lipid molecule, promote lysosomal degradation? Upstream of the lysosome, autophagosome formation is mediated by a lipidation of LC3 by ATG3 (*Ichimura et al., 2000*). Thus, we hypothesized that LOOH may affect the lipidation of LC3. Indeed, GPx4 KD drastically reduced the protein content of p62, LC3-I, and LC3-II (*Figure 5C* and *Figure 5—figure supplement 1D–H*), potentially suggesting that LOOH either accelerates lysosomal degradation or reduces synthesis of these proteins. To test this possibility, we performed a targeted

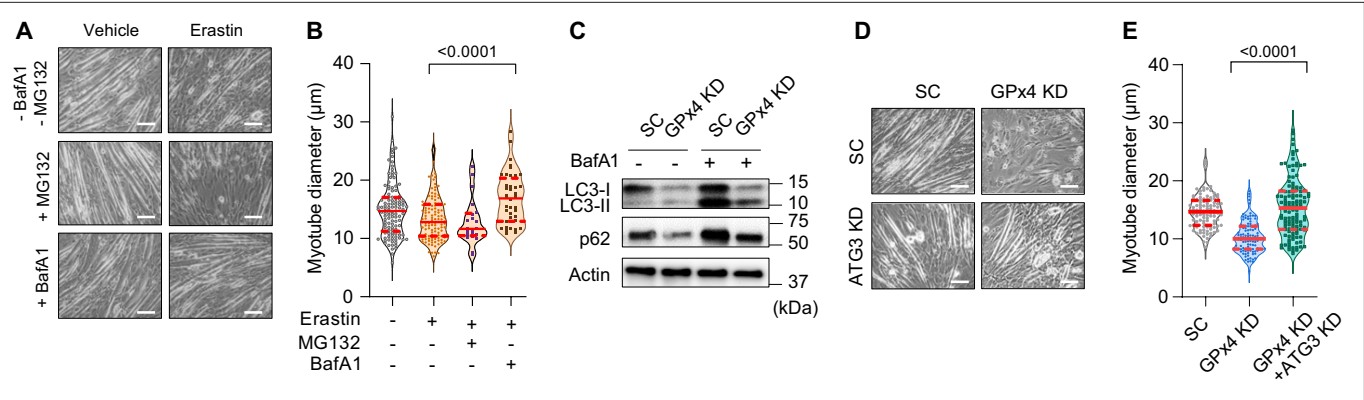

**Figure 5.** Suppression of the autophagy-lysosome axis prevents lipid hydroperoxides (LOOH)-induced myotube atrophy. (**A, B**) Representative images (**A**), and quantification of myotube diameter (**B**) from erastin-stimulated C2C12 myotubes with proteasomal inhibitor MG132 or lysosomal inhibitor bafilomycin A1 (BafA1) (n=21–120 per group). Scale bar, 100 µm. (**C**) Autophagic flux analyses immunoblotting for LC3-I, LC3-II, p62, and actin in SC or glutathione peroxidase 4 (GPx4) knockdown (KD) C2C12 myotubes with or without BafA1. (**D, E**) Representative images (**D**) and quantification of myotube diameter (**E**) from C2C12 myotubes with or without GPx4 KD and/or ATG3 KD (n=104–121 per group). Data are shown as the mean ± SEM. Statistical analyses in (**B, E**) were performed with a one-way analysis of variance (ANOVA) with Dunnett's multiple comparisons test.

The online version of this article includes the following source data and figure supplement(s) for figure 5:

**Source data 1.** Original western blots.

**Figure supplement 1.** The autophagy-lysosome axis in lipid hydroperoxides (LOOH)-induced myotube atrophy.

**Figure supplement 1—source data 1.** Original western blots.

deletion of ATG3 in vitro. Indeed, ATG3 KD completely rescued the reduction in myotube diameter induced by GPx4 KD (*Figure 5D, E* and *Figure 5—figure supplement 1I–L*).

Leveraging these findings, we generated mice with skeletal muscle-specific tamoxifen-inducible ATG3 knockout (ATG3-MKO) (*Figure 6A, B* and *Figure 6—figure supplement 1A*) and studied them with or without HU (*Figure 6—figure supplement 1B–E*). Loss of ATG3 was protective from disuse-induced atrophy (*Figure 6C* and *Figure 6—figure supplement 1F*) in soleus and weakness (*Figure 6D* and *Figure 6—figure supplement 1G*) in soleus and EDL. The protective effect of muscle mass in soleus was likely explained by greater myofiber CSA (*Figure 6E, F* and *Figure 6—figure supplement 1H*). Thus, suppression of autophagy is sufficient to attenuate disuse-induced muscle atrophy and weakness.

We initially hypothesized that lysosomal degradation mediates LOOH-induced protein degradation to contribute to muscle atrophy. However, further assessment of muscle LOOH illuminated a more complex interaction between the lysosome and LOOH (*Chen et al., 2021*; *Gao et al., 2016*). Unexpectedly, quantification of 4-HNE revealed that inhibition of the autophagy-lysosome axis by ATG3 deletion or BafA1 was sufficient to inhibit LOOH induced by GPx4 deletion, erastin, or RSL3 (*Figure 7A–D* and *Figure 7—figure supplement 1A–D*). These findings suggest that the autophagy-lysosome axis is essential for LOOH amplification, in addition to its potential role in mediating protein degradation downstream. Indeed, immunofluorescence experiments revealed that 4-HNE is highly co-localized to LAMP2 (*Figure 7E* and *Figure 7—figure supplement 1E*), consistent with the notion that the lysosome is necessary for LOOH propagation. To support this idea, an increase in LOOH by hydrogen peroxide or carmustine (agents that increase pan oxidative stress without acting on GPx4 directly) was completely inhibited by lysosomal inhibition (*Figure 7F–H* and *Figure 7—figure supplement 1F, G*). Together, these observations suggest that the propagation of LOOH may be mediated by the lysosome (*Figure 7—figure supplement 1H*).

Inhibition of autophagy suppressed lysosomal degradation and LOOH to attenuate muscle atrophy. We next tested whether suppression of LOOH would be sufficient to ameliorate skeletal muscle atrophy. We studied young (4 months) and old (20 months) global GPx4-overexpressing (GPx4Tg) mice (*Ran et al., 2004*) with or without HU (*Figure 8—figure supplement 1A–E*). Strikingly, both young and old GPx4Tg mice were resistant to disuse-induced muscle atrophy (*Figure 8A* and *Figure 8—figure supplement 1F, G*, greater effect in soleus than EDL). Perhaps even more impactful

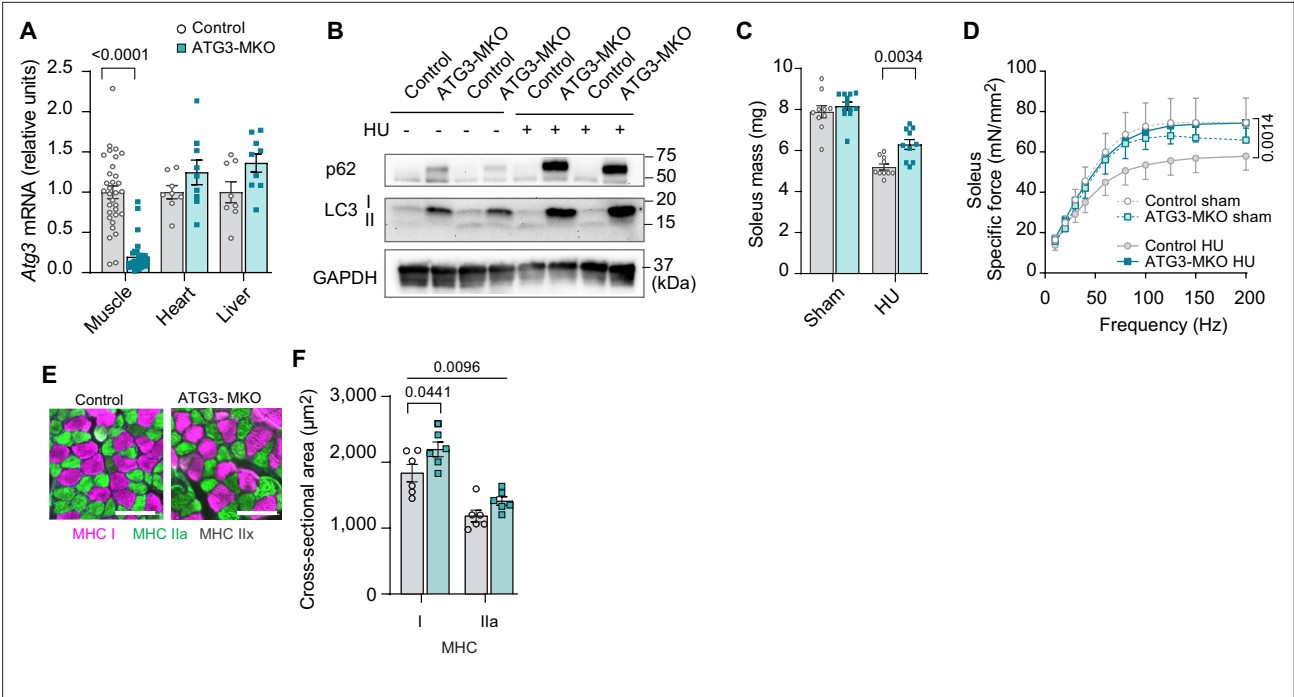

**Figure 6.** Muscle-specific ATG3 deletion attenuates disuse-induced atrophy and weakness. (**A**) mRNA levels of *Atg3* (n=8–32 per group) from young control or ATG3-MKO mice. (**B**) Immunoblotting of p62, LC3, and GAPDH proteins from gastrocnemius muscles from control and ATG3-MKO mice. (**C**) Soleus muscle mass from control or ATG3-MKO mice (n=10–12 per group). (**D**) Force-frequency curve from young control or ATG3-MKO mice (n=9–12 per group). (**E, F**) Representative images of MHC immunofluorescence (**E**) and muscle fiber cross-sectional area (CSA) by fiber type (**F**) of soleus muscles in young control or ATG3-MKO mice with hindlimb unloading (HU) (n=6 per group). Data from ATG3-MKO experiments are from female mice. Data are shown as the mean ± SEM. Statistical analyses in (**A**) were performed with an unpaired two-tailed t-test. Statistical analyses in (**C, D**) and (**F**) were performed with a two-way analysis of variance (ANOVA) and multiple comparisons were performed using Tukey's multiple comparisons tests.

The online version of this article includes the following source data and figure supplement(s) for figure 6:

**Source data 1.** Original western blots.

**Figure supplement 1.** Additional data from ATG3-MKO mice.

**Figure supplement 1—source data 1.** Original genotyping gels and western blots.

was the effect of GPx4 overexpression on skeletal muscle force-generating capacity such that, in both young and old, GPx4 overexpression robustly protected mice from muscle weakness induced by HU (*Figure 8B* and *Figure 8—figure supplement 1H–J*, soleus and EDL). These findings are in contrast to our experiments in *Gpx4*[+/-] mice where muscle mass phenotype was only present in the young mice (*Figure 3A*) and no phenotype on muscle strength (*Figure 3C*). Consistent with the notion that GPx4 overexpression acts on LOOH, HU-induced increase in 4-HNE was completely suppressed in GPx4Tg mice (*Figure 8C* and *Figure 8—figure supplement 1K*). We also found that the protection from muscle atrophy was explained by greater myofiber CSA regardless of fiber type (*Figure 8D&E* and *Figure 8—figure supplement 1L, M*).

Next, we explored opportunities to pharmacologically suppress LOOH to prevent muscle atrophy. Ferrostatin-1 inhibits the propagation of lipid peroxidation and is widely used to study LOOH (*Dixon et al., 2012*; *Codenotti et al., 2018*). Indeed, incubation of cells with ferrostatin-1 was sufficient to suppress LOOH induced by GPx4 KD (*Figure 8F* and *Figure 8—figure supplement 2A*) concomitant with protection from myotube atrophy (*Figure 8G&H* and *Figure 8—figure supplement 2B, C*). Nevertheless, ferrostatin-1 is currently not an FDA-approved drug with uncertainty surrounding safety. Thus, we tested L-carnosine, a dipeptide composed of beta-alanine and L-histidine that has the ability to scavenge reactive lipid aldehydes formed from LOOH (*Cripps et al., 2017*; *Everaert et al., 2013*). Rather than acting to suppress the lipid peroxidation process, L-carnosine binds to reactive lipid aldehydes to neutralize carbonyl stress. Similar to ferrostatin-1, L-carnosine was sufficient to suppress 4-HNE and rescue cell death induced by GPx4 KD (*Figure 8—figure supplement 2D–F*) or erastin

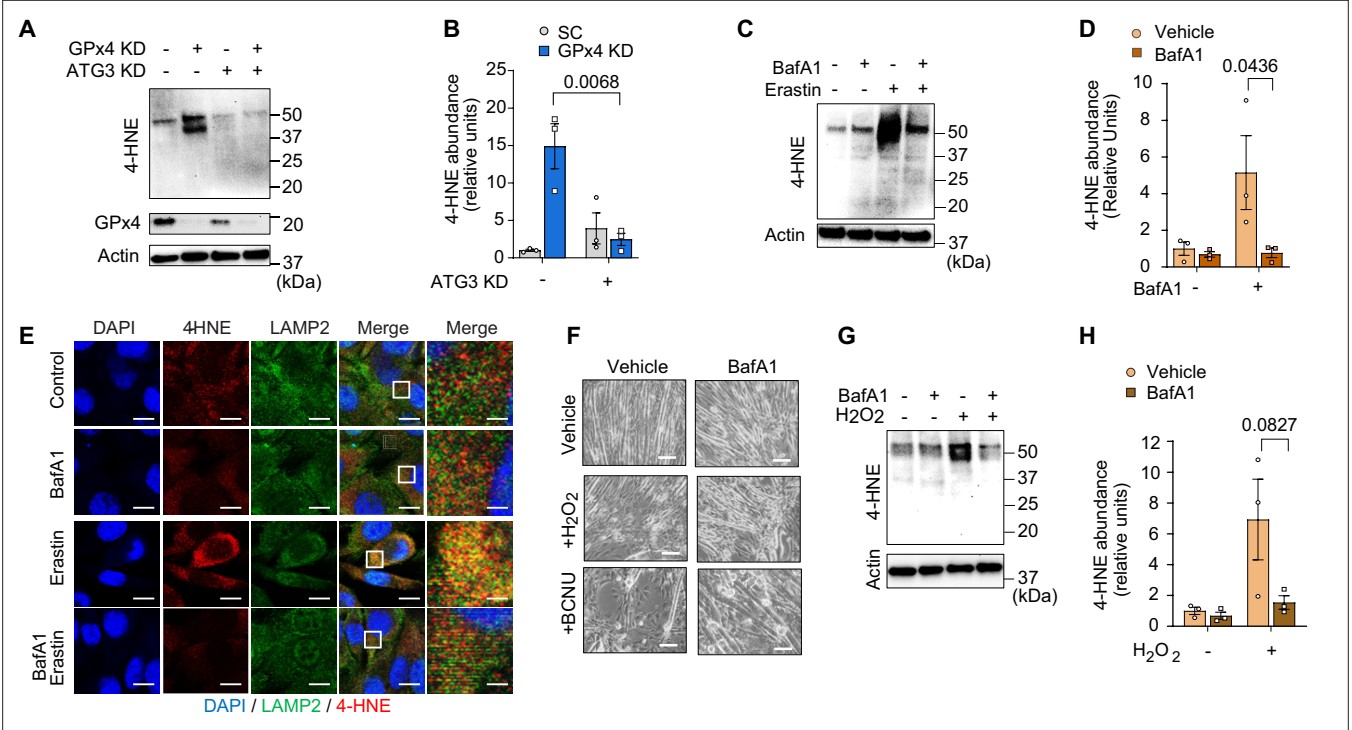

**Figure 7.** Inhibition of the autophagy-lysosome axis prevents accumulation of lipid hydroperoxides (LOOH). (**A, B**) Immunoblotting (**A**) and quantification (**B**) of 4-hydroxynonenal (4-HNE), glutathione peroxidase 4 (GPx4), and actin in C2C12 myotubes with or without GPx4 knockdown (KD) and/or ATG3 KD (n=3 per group). (**C, D**) Immunoblotting (**C**) and quantification (**D**) of 4-HNE protein from C2C12 myotubes with or without erastin and/or bafilomycin A1 (BaFA1) (n=3 per group). (**E**) Confocal fluorescence microscope images of erastin-stimulated myotubes with or without BaFA1. Scale bar, 10 µm. Boxed regions are shown enlarged at far right. Scale bar, 2.5 µm. (**F, G, H**) Representative images (**F**), immunoblotting of 4-HNE and actin (**G**), and quantification (**H**) of $H_2O_2$ or BCNU-stimulated C2C12 myotubes with or without BaFA1 (n=3 per group). Scale bar, 100 µm. Data are shown as the mean ± SEM. Statistical analyses in (**B, D**) and (**H**) were performed with a two-way analysis of variance (ANOVA) and Tukey's multiple comparison test.

The online version of this article includes the following source data and figure supplement(s) for figure 7:

**Source data 1.** Original western blots.

**Figure supplement 1.** Inhibition of the autophagy-lysosome axis prevents accumulation of lipid hydroperoxides (LOOH).

**Figure supplement 1—source data 1.** Original western blots.

---

(*Figure 8—figure supplement 2G–I*). Leveraging these data, we performed a preclinical trial for L-carnosine provided in drinking water ad lib (80 mM) in young wildtype C57BL6/J mice. L-carnosine treatment did not alter body mass, body composition, food intake, and water intake (*Figure 8—figure supplement 3A–D*), and successfully suppressed 4-HNE induced by HU (*Figure 8—figure supplement 3E, F*). Remarkably, mice provided with L-carnosine were partly protected from disuse-induced atrophy in soleus muscles (*Figure 8—figure supplement 3G*).

In humans, L-carnosine is rapidly degraded by a circulating carnosinase (*Boldyrev et al., 2013*) that may render oral carnosine treatment ineffective. In contrast, *N*-acetylcarnosine has a longer half-life and may be a more effective reagent in humans thus improving its translational potential. Similar to ferrostatin-1 and L-carnosine, *N*-acetylcarnosine also prevented 4-HNE and cell death induced by GPx4 KD (*Figure 9—figure supplement 1A–C*) or erastin (*Figure 9—figure supplement 1D–F*). Thus, we proceeded with a preclinical trial for *N*-acetylcarnosine in drinking water (80 mM, *Figure 9A*) in young (4 months, C57BL6/J; Jax colony) and old (20 months, C57BL/6; NIA rodent colony) wildtype mice. Similar to L-carnosine treatment, *N*-acetylcarnosine did not alter body mass, body composition, food intake, or water intake (*Figure 9—figure supplement 2A–E*), and successfully suppressed muscle 4-HNE (*Figure 9B* and *Figure 9—figure supplement 2F, G*). Strikingly, similar to our findings in GPx4Tg mice, *N*-acetylcarnosine ameliorated atrophy in soleus (*Figure 9C* and *Figure 9—figure supplement 2H*) and weakness in both soleus and EDL (*Figure 9D* and *Figure 9—figure supplement*

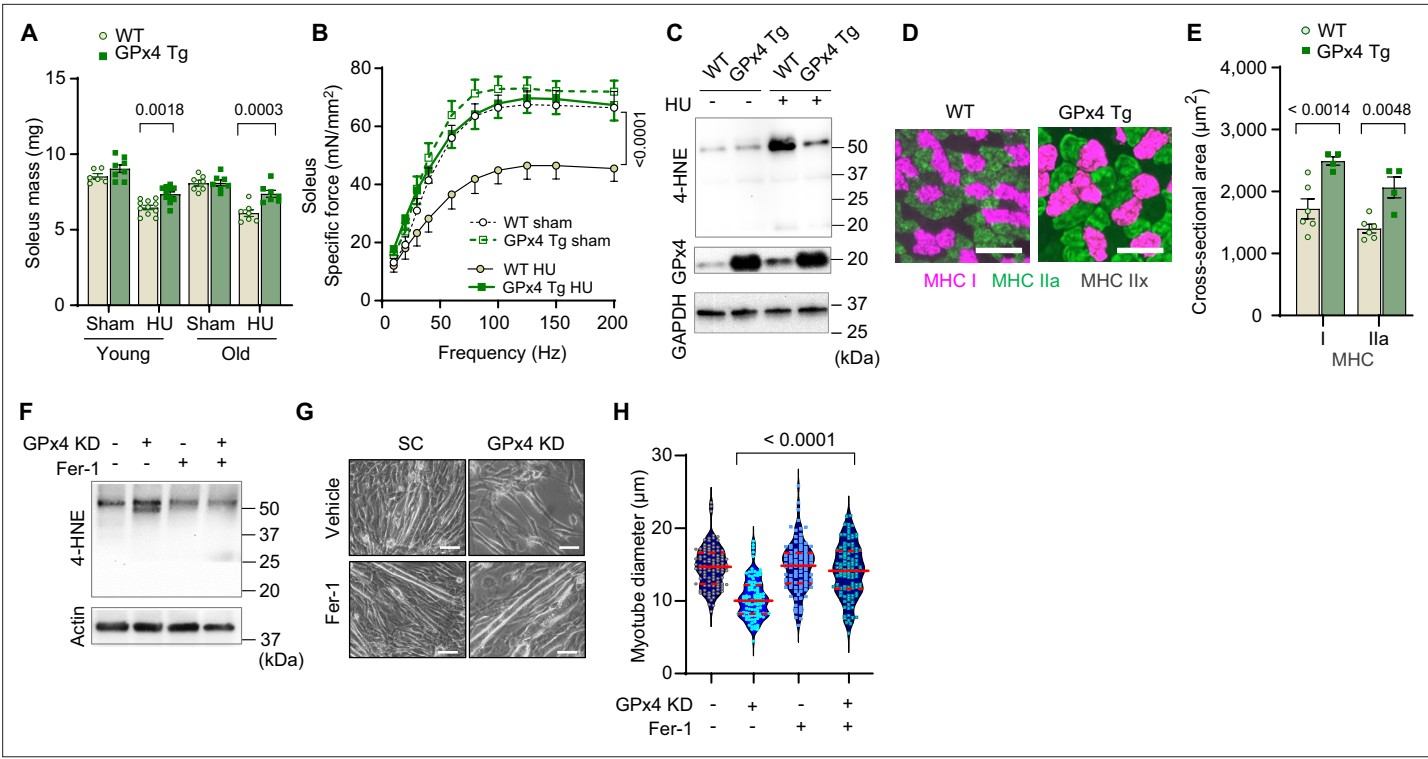

**Figure 8.** Overexpression of glutathione peroxidase 4 (GPx4) ameliorates disuse-induced muscle atrophy and weakness in young and old mice. (**A**) Soleus muscle mass from young or old wildtype (WT) or GPx4Tg mice with or without hindlimb unloading (HU) (n=6–11 per young group, n=7 per old group). (**B**) Force-frequency curve from old WT or GPx4Tg mice (n=5–7 per group). (**C**) Immunoblotting of 4-hydroxynonenal (4-HNE) in gastrocnemius muscles from old WT or GPx4Tg mice. (**D, E**) Representative images of MHC immunofluorescence (**D**) and muscle fiber cross-sectional area (CSA) by fiber type (**E**) for soleus muscles in old WT or GPx4Tg mice with HU (n=4–6 per group). (**F**) Immunoblotting of 4-HNE, and actin in C2C12 myotubes with or without GPx4 KD and/or ferrostatin-1. (**G, H**) Representative images (**G**), and quantification of myotube diameter (**H**) from C2C12 myotubes with GPx4 KD and/or ferrostatin-1 treatments (n=102–110 per group). Scale bar, 100 μm. Data from GPx4Tg experiments are from male and female mice. Data are shown as the mean ± SEM. Statistical analyses in (**H**) was performed with a one-way analysis of variance (ANOVA) with Dunnett's multiple comparisons test, and statistical analyses in (**A, B**) and (**E**) were performed with a two-way ANOVA and were performed using Tukey's (**B, E**) or Sidak's (**A**) multiple comparisons tests.

The online version of this article includes the following source data and figure supplement(s) for figure 8:

**Source data 1.** Original western blots.

**Figure supplement 1.** Additional data from GPx4Tg mice.

**Figure supplement 2.** Effects of ferrostatin-1 or L-carnosine treatment in vitro.

**Figure supplement 2—source data 1.** Original western blots.

**Figure supplement 3.** L-carnosine treatment partly ameliorates disuse-induced muscle atrophy.

**Figure supplement 3—source data 1.** Original blots.

*2I–K*) in both young and old mice. Protection from muscle atrophy was similarly explained by greater myofiber CSA regardless of fiber type (*Figure 9E, F* and *Figure 9—figure supplement 2L, M*).

## Discussion

The current findings demonstrate a novel mechanism that indicate LOOH as a key downstream molecule by which oxidative stress promotes muscle atrophy and weakness. Skeletal muscle LOOH was robustly upregulated with aging and disuse, and genetic or pharmacological neutralization of LOOH and their secondary reactive lipid aldehydes was sufficient to rescue muscle atrophy and weakness. In particular, *N*-acetylcarnosine treatment shows a potent effect in preserving muscle mass and strength with disuse in both young and old mice, informing the potential trial to utilize this compound to ameliorate loss of muscle function in humans.

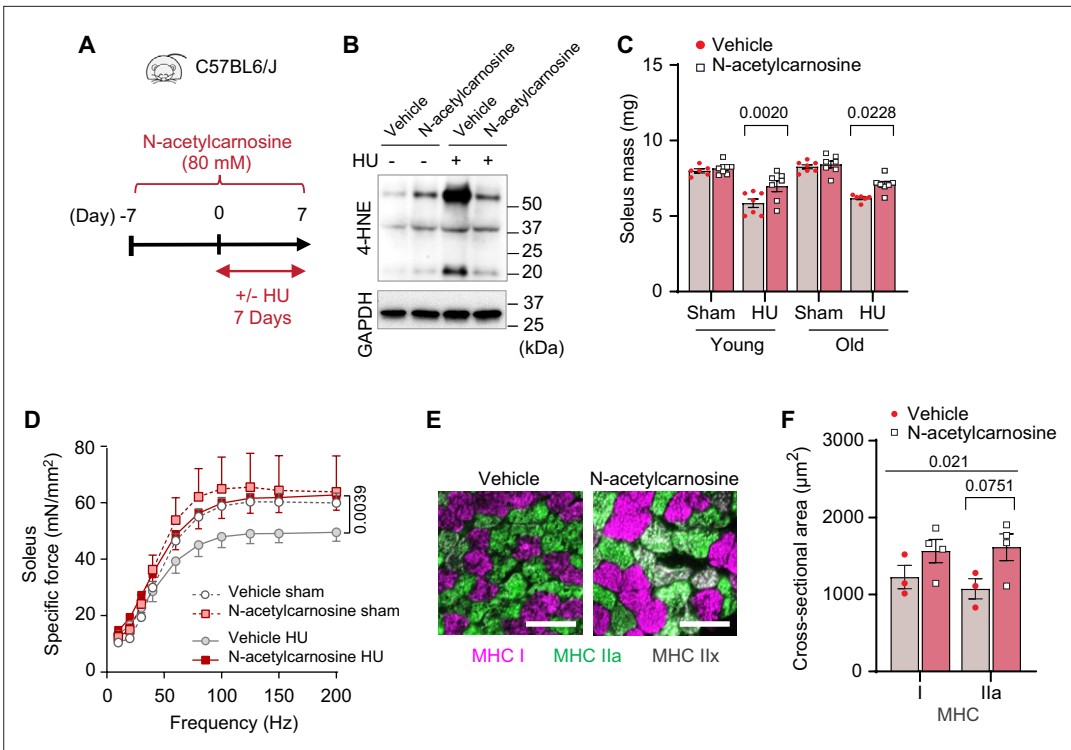

**Figure 9.** Pharmacological suppression of carbonyl stress ameliorates muscle atrophy and weakness in young and old mice. (**A**) Schematic illustration of the protocol for administration of *N*-acetylcarnosine in vivo. (**B**) Immunoblotting of 4-hydroxynonenal (4-HNE) in gastrocnemius muscles from *N*-acetylcarnosine treatment in old mice. (**C**) Soleus muscle mass from young or old mice with or without *N*-acetylcarnosine treatment (n=6–8 per young group, n=7 per old group). (**D**) Force-frequency curve from *N*-acetylcarnosine study in old mice (n=4–5 per group). (**E, F**) Representative images of MHC immunofluorescence (**E**) and muscle fiber cross-sectional area (CSA) by fiber type (**F**) for soleus muscles (n=3–4 per group) in old mice from the *N*-acetylcarnosine study. Data from *N*-acetylcarnosine experiments are from male and female mice. Data are shown as the mean ± SEM. Statistical analyses in (**C, D**) and (**F**) were performed with a two-way analysis of variance (ANOVA) and were performed using Tukey's (**C, F**) or Sidak's (**D**) multiple comparisons tests.

The online version of this article includes the following source data and figure supplement(s) for figure 9:

**Source data 1.** Original western blots.

**Figure supplement 1.** Effects of *N*-acetylcarnosine treatment in vitro.

**Figure supplement 1—source data 1.** Original western blots.

**Figure supplement 2.** Additional data from *N*-acetylcarnosine treatment in vivo.

**Figure supplement 2—source data 1.** Original blots.

During the preparation of this manuscript, Van Remmen and colleagues published a complementary study demonstrating that liproxstatin-1 can suppress denervation-induced skeletal muscle atrophy (***Brown et al., 2022***). Denervation and HU elicits different but overlapping response in myofibers, and our studies demonstrate that their effects to drive skeletal muscle atrophy might converge on lipid peroxidation. Like ferrostatin-1, liproxstatin-1 acts to suppress the propagation of lipid peroxidation rather than acting directly on LOOH. Nevertheless, in vivo liproxstatin-1 treatment was highly effective in suppressing denervation-induced LOOH as well as reactive lipid aldehydes 4-HNE, suggesting that targeting lipid peroxidation is likely an equally effective strategy to suppress LOOH production in skeletal muscle. Conversely, our data with GPx4 overexpression and *N*-acetylcarnosine treatment indicate that the effect of lipid peroxidation to promote muscle atrophy is mediated by LOOH and their lipid reactive aldehydes. Neither liproxstatin-1 nor ferrostatin-1 are currently FDA-approved, but it is worthwhile to consider these drugs along with *N*-acetylcarnosine as potential therapeutics to treat disuse-induced muscle atrophy.

Inhibition of the autophagy-lysosomal axis prevented myotube atrophy induced by erastin or GPx4 deletion, and muscle atrophy and weakness induced by HU in vivo. These findings are in contrast with previous studies that have shown inhibition of autophagy accelerates muscle mass (*Carnio et al., 2014*; *Castets et al., 2013*; *Masiero et al., 2009*). We do not interpret these differences as a discrepancy. In our studies, both inhibition of autophagy-lysosomal axis and induction of LOOH (in vitro or HU) were performed short term. It is likely that ATG3-MKO mice will eventually begin to undergo muscle atrophy similar to the muscle-specific ATG7 knockout model (*Masiero et al., 2009*). Importantly, inhibition of the autophagy-lysosomal axis not only inhibited muscle atrophy but also the accumulation of LOOH. The exact mechanism of the role of the lysosome in LOOH propagation is unclear, but our preliminary evidence shows LOOH accumulation occurring near the lysosome. Further studies are needed to clarify the exact nature of this interaction.

Sarcopenia is an age-associated decline in muscle mass and strength, that occurs due to a combination of inactivity, injury, and/or surgery, in addition to the biological consequences of aging itself. In the current study, mice were studied at 4 or 20 months of age. While not statistically compared directly (these experiments were not performed side-by-side), skeletal muscle mass at 20 months of age was not significantly lower compared to those at 4 months of age. Thus, the current data is unclear whether genetic or pharmacological suppression of LOOH prevents the loss of muscle mass due to the biological effect of aging in the absence of HU. We chose to study mice with 20 months of age for two reasons. First, mice greater than 20 months of age do not tolerate the HU intervention well, often resulting in their inability to consume food or water. Because disuse is an integral component of human aging, we wanted to study how muscles from old mice respond to disuse. This therefore compromised our ability to study sarcopenia without disuse. Second, while muscle mass was not diminished at 20 months of age, skeletal muscle force-generating capacity was lower in 20-month-old mice compared to the 4-month-old mice, particularly in the extensor digitorum longus (EDL) muscles. In GPx4Tg mice, age-associated decrease in muscle strength (in non-HU mice) appeared to be rescued, while short-term treatment with *N*-acetylcarnosine had no effect. We are currently following up on these results with a long-term treatment of *N*-acetylcarnosine with age that display the loss of muscle mass independent of HU to see if such intervention might alleviate the loss of muscle mass and strength due to the effect of aging per se. Studies in mice greater than 20 months of age would be necessary to more definitively show the role of LOOH in sarcopenia.

We initially set out to investigate the role of LOOH in age- and disuse-induced skeletal muscle atrophy, while measuring force-generating capacity as a secondary outcome. However, in all experimental models in which accumulation of muscle LOOH was suppressed (young and old GPx4Tg mice, young and old mice with *N*-acetylcarnosine, and young ATG3-MKO mice), force-generating capacity (i.e., specific force normalized to CSA) was more robustly rescued compared to skeletal muscle mass. This suggests the role of LOOH to induce muscle weakness independent of muscle atrophy, and likely independent of muscle protein degradation. While out of the scope for the current study, it would be important to determine whether reactive lipid aldehydes induced by aging or disuse preferentially bind to enzymes of skeletal muscle contraction or excitation-contraction coupling to compromise their activities. As described in the previous paragraph, aging promoted muscle weakness prior to atrophy. Similarly, muscle atrophy induced by cancer cachexia is also preceded by muscle weakness (*Delfinis et al., 2022*). Thus, these observations highlight the need to better study the mechanisms that regulate force-generating capacity independent of muscle mass.

In conclusion, we provide evidence that LOOH contributes to the loss of muscle mass and strength associated with age and disuse. Neutralization of LOOH, particularly their reactive lipid aldehyde byproducts, attenuates muscle atrophy and weakness. The mechanisms by which LOOH contributes to these phenotypes are not entirely clear, but they include protein degradation mediated by the autophagy-lysosomal axis, as well as loss in the force-generating capacity that is likely mediated by carbonyl stress. Last, but not least, these promising observations inform a potential clinical trial to test the efficacy of *N*-acetylcarnosine treatment in ameliorating muscle atrophy in humans.

# Methods

**Key resources table**

| Reagent type (species) or resource | Designation | Source or reference | Identifiers | Additional information |
|---|---|---|---|---|
| Antibody | 4-Hydroxynonenal (4-HNE) (Mouse monoclonal) | Abcam | Ab48506 | WB (1:1000) |
| Antibody | Actin (Rabbit Polyclonal) | MilliporeSigma | A2066 | WB (1:1000) |
| Antibody | Alexa Fluor 647-conjugated secondary (Goat Anti-Mouse Polyclonal) | Invitrogen | A21242 | IF (1:250) |
| Antibody | Alexa Fluor 568-conjugated secondary (Donkey Anti-Mouse Polyclonal) | Abcam | Ab175472 | IF (1:500) |
| Antibody | Alexa Fluor 555-conjugated secondary (Goat Anti-Mouse Polyclonal) | Invitrogen | A21426 | IF (1:500) |
| Antibody | Alexa Fluor 488-conjugated secondary (Donkey Anti-Rabbit Polyclonal) | Abcam | Ab150073 | IF (1:500) |
| Antibody | Alexa Fluor 488-conjugated secondary (Goat Anti-Mouse Polyclonal) | Invitrogen | A21121 | IF (1:500) |
| Antibody | GAPDH (Rabbit Monoclonal) | Cell Signaling Technology | 14C10 | WB (1:1000) |
| Antibody | GPx4 (Rabbit Monoclonal) | Abcam | Ab125066 | WB (1:1000) |
| Antibody | LAMP-2 (Rabbit Polyclonal) | Novus | NB300-591 | IF (1:200) |
| Antibody | LC3B (Mouse Monoclonal) | Cell Signaling Technology | 83506 | WB (1:1000) |
| Antibody | Myosin Heavy Chain Type I (Mouse Monoclonal) | DSHB | BA.D5 | IF (1:100) |
| Antibody | Myosin Heavy Chain Type IIA (Mouse Monoclonal) | DSHB | SC.71 | IF (1:100) |
| Antibody | Myosin Heavy Chain Type IIB (Mouse Monoclonal) | DSHB | BF.F3 | IF (1:100) |
| Antibody | P62 (Mouse Monoclonal) | Abcam | Ab56416 | WB (1:1000) |
| Biological Sample (Human) | Human Muscle Biopsy Samples | *Tanner et al., 2015* *Reidy et al., 2017* | N/A | |
| Cell Line, (*Mus musculus*) | C2C12 Myoblasts | ATCC | CRL-1772 | |
| Cell Line, (Human) | HEK293T | ATCC | CTRL-3216 | |
| Strain, strain background (*Mus musculus*) | C57BL/6J; Wild Type (WT) | The Jackson Laboratory | 000664 | Male and Female |
| Strain, strain background (*Mus musculus*) | GPx4 heterogeneous KO (*Gpx4*+/-) | *Yant et al., 2003*. | N/A | |
| Strain, strain background (*Mus musculus*) | GPx4 overexpression (GPx4Tg) | *Ran et al., 2004*. | N/A | |
| Strain, strain background (*Mus musculus*) | GPx4 conditional KO (GPx4 cKO lox/lox) | The Jackson Laboratory | 027964 | |

*Continued on next page*

*Continued*

| Reagent type (species) or resource | Designation | Source or reference | Identifiers | Additional information |
|---|---|---|---|---|
| Strain, strain background (*Mus musculus*) | ATG3 conditional KO (ATG3 cKO lox/lox) | *Cai et al., 2018*. | N/A | |
| Strain, strain background (*Mus musculus*) | HSA-MerCreMer$^{+/-}$ | *McCarthy et al., 2012*. | N/A | |
| Strain, strain background (*Mus musculus*) | *Gpx4 sh*RNA | MilliporeSigma | TRCN0000076552 | |
| Strain, strain background (*Mus musculus*) | *Lpcat3 sh*RNA | MilliporeSigma | TRCN0000121437 | |
| Strain, strain background (*Mus musculus*) | *Atg3 sh*RNA | MilliporeSigma | TRCN0000247442 | |
| Recombinant DNA Reagent | Packaging Vector psPAX2 | Addgene | 12260 | |
| Recombinant DNA Reagent | Envelope Vector pMD2.G | Addgene | 12259 | |
| Sequence-based reagent | Scrambled shRNA plasmid | Addgene | 1864 | |
| Sequence-based reagent | Mouse *Gpx4* Fwd Primer | U of U Genomics Core | GCTGAGAATTCGTGCATGG | |
| Sequence-based reagent | Mouse *Gpx4* Rev Primer | U of U Genomics Core | CCGTCTGAGCCGCTTACTTA | |
| Sequence-based reagent | Mouse *Atg3* Fwd Primer | U of U Genomics Core | ACACGGTGAAGGGAAAGGC | |
| Sequence-based reagent | Mouse *Atg3* Rev Primer | U of U Genomics Core | TGGTGGACTAAGTGATCTCCAG | |
| Sequence-based reagent | Mouse *Chac1* Fwd Primer | U of U Genomics Core | CTGTGGATTTTCGGGTACGG | |
| Sequence-based reagent | Mouse *Chac1* Rev Primer | U of U Genomics Core | CCCCTATGGAAGGTGTCTCC | |
| Sequence-based reagent | Mouse *Ptgs2* Fwd Primer | U of U Genomics Core | TGAGCAACTATTCCAAACCAGC | |
| Sequence-based reagent | Mouse *Ptgs2* Rev Primer | U of U Genomics Core | GCACGTAGTCTTCGATCACTATC | |
| Sequence-based reagent | Mouse *Lpcat3* Fwd Primer | U of U Genomics Core | GGCCTCTCAATTGCTTATTTCA | |
| Sequence-based reagent | Mouse *Lpcat3* Rev Primer | U of U Genomics Core | AGCACGACACATAGCAAGGA | |
| Chemical compound, drug | Auranofin | Sigma-Aldrich | A6733 | |
| Chemical compound, drug | BaFA1 | MilliporeSigma | SML1661 | |
| Chemical compound, drug | Erastin | MilliporeSigma | E7781 | |
| Chemical compound, drug | Ferrostatin-1 | MilliporeSigma | SML0583 | |
| Chemical compound, drug | L-carnosine | MilliporeSigma | C9625 | |
| Chemical compound, drug | *N*-acetylcarnosine | Cayman Chemical | 18817 | |
| Chemical compound, drug | RSL3 | MilliporeSigma | SLM2234 | |
| Commercial assay or kit | MDA Lipid Peroxidation Assay | Abcam | Ab118970 | |
| Software, algorithm | GraphPad Prism 9.3 | GraphPad | N/A | |
| Software, algorithm | ImageJ | NIH | N/A | |

## Animal models

*Gpx4$^{+/-}$* and GPx4Tg mice were generated previously (*Yant et al., 2003*; *Ran et al., 2004*). Conditional GPx4 knockout (GPx4cKO lox/lox) mice were acquired from Jackson Laboratory (Stock No: 027964) (*Yoo et al., 2012*). Conditional ATG3 knockout (ATG3cKO lox/lox) mice were previously described (*Cai et al., 2018*). GPx4cKO lox/lox mice or ATG3cKO lox/lox mice were then crossed with tamoxifen-inducible, skeletal muscle-specific Cre recombinase (HSA-MerCreMer$^{+/-}$) mice (*McCarthy et al., 2012*) to generate GPx4cKO lox/lox; HSAMerCreMer$^{-/-}$ (control) and GPx4cKO lox/lox; HSA-MerCreMer$^{+/-}$ (skeletal muscle-specific GPx4 knockout; GPx4-MKO) mice or ATG3cKO lox/lox;

HSAMerCreMer[-/-] (control) and ATG3cKO lox/lox; HSA-MerCreMer[+/-] (ATG3-MKO) mice. Tamoxifen-injected (7.5 µg/g body mass, 5 consecutive days) littermates were used. Mice were maintained on a 12 hr light/12 hr dark cycle in a temperature-controlled room. Body composition measurements were taken immediately before terminal experiments with a Bruker Minispec MQ20 nuclear magnetic resonance analyzer (Bruker, Rheinstetten, Germany). All mice were bred onto C57BL/6J background and were born at normal Mendelian ratios. Body mass were measured every day during HU. All experiments were randomized and blinded where appropriate. No data were excluded from the study and ARRIVE guidelines 2.0 has been followed. All protocols were approved by Institutional Animal Care and Use Committees at the University of Utah (#20-07007).

## Human skeletal muscle samples

Skeletal muscle biopsy samples from human bedrest studies were collected from a previous study (*Mahmassani et al., 2019*). Informed consent and consent to publish was obtained from subjects. The study was reviewed and approved by the University of Utah Institutional Review Board and conformed to the Declaration of Helsinki and Title 45, US Code of Federal Regulations, Part 46, 'Protection of Human Subjects'.

## Hindlimb unloading

Mice underwent 1, 7, or 14 days of HU (2 mice/cage) using a previously described protocol (*Eshima et al., 2020*; *Heden et al., 2019*) based on the traditional Morey-Holton design to study disuse atrophy in rodents. Along with daily monitoring of body mass, food intake was monitored every other day to ensure that the mice did not experience excessive weight loss due to malnutrition or dehydration. Following 1, 7, or 14 days of HU, mice were fasted for 4 hr (to avoid mice feeding immediately prior to terminal experiments) and given an intraperitoneal injection of 80 mg/kg ketamine and 10 mg/kg xylazine, after which tissues were harvested. EDL and soleus were carefully dissected for weight measurements.

## Muscle force generation

Force-generating properties of soleus and EDL muscles were measured as previously described (*Ferrara et al., 2018*; *Verkerke et al., 2019*). Briefly, soleus/EDL muscles were sutured at each tendon, and muscles were suspended at optimal length (Lo), which was determined by pulse stimulation. After Lo was identified, muscles were stimulated (0.35 s, pulse width 0.2 ms) at frequencies ranging from 10 to 200 Hz. Muscle length and mass were measured to quantify CSA for force normalization.

## Quantitative reverse transcription PCR

Samples were homogenized in TRIzol reagent (Life Technologies) to extract total RNA. One microgram RNA was reverse-transcribed using an IScript cDNA synthesis kit (Bio-Rad). Reverse transcription PCR was performed with the Viia 7 Real-Time PCR System (Life Technologies) using SYBR Green reagent (Life Technologies). All data were normalized to ribosomal L32 gene expression and were normalized to the mean of the control group. Primers were based on sequences in public databases.

## Western blot

Whole muscle or cells were homogenized, and western blots were performed as previously described (*Eshima et al., 2020*). Protein homogenates were analyzed for abundance of phosphorylated 4-HNE (ab48506; Abcam), GPx4 (ab125066, Abcam), actin (A2066, MilliporeSigma), GAPDH (14C10, Cell Signaling Technology), p62 (ab56416, Abcam), LC3B (83506, Cell Signaling Technology).

## Mass spectrometry

Oxidolipidomics samples were analyzed on the SCIEX 7500 system coupled with ExionLC (SCIEX, Concord, Canada) using multiple reaction monitoring analysis. Mobile phase A is composed of 93:7 acetonitrile:dichloromethane containing 2 mM ammonium acetate and mobile phase B is composed of 50:50 acetonitrile:water containing 2 mM ammonium acetate. A Phenomenex Luna NH2 column with 3 µm particle size (4.6×150 mm$^2$) was used for separation and column temperature was kept at 40°C. The total flow rate is 0.7 mL/min with a total run time of 17 min. Samples were extracted using

the Bligh and Dyer method. Lower layer was collected, dried down, and resuspended in mobile phase A.

## Cell culture

C2C12 myoblasts and HEK293T cells were obtained and authenticated by ATCC, and tested negative for mycoplasma contamination. Both cell lines were grown and maintained in high-glucose Dulbecco's modified Eagle's medium (DMEM), with 10% fetal bovine serum, and 0.1% penicillin/streptomycin. Once 90–100% confluent, C2C12 cells were differentiated into myotubes with low-glucose DMEM, with L-glutamine and 110 mg/L sodium pyruvate, supplemented with 2% horse serum, and 0.1% penicillin-streptomycin. For experiments with erastin (E7781, MilliporeSigma), Ferrostatin-1 (SML0583, MilliporeSigma), and RSL3 (SML2234, MilliporeSigma), C2C12 myotubes were incubated with either 10 μM erastin/10 μM ferrostatin-1/5 μM RSL3/or equal-volume DMSO directly dissolved into medium. For experiments with L-carnosine (C9625, MilliporeSigma), and N-acetylcarnosine (18817, Cayman), C2C12 myotubes were incubated with 10 mM of L-carnosine/N-acetylcarnosine directly dissolved into medium.

## Lentivirus-mediated KD of GPx4/LPCAT3/ATG3

Lentivirus-mediated KD of experiments were performed as previously described (*Heden et al., 2019*; *Ferrara et al., 2021a*; *Johnson et al., 2020*). Vectors were decreased using pLKO.1 lentiviral-RNAi system. Plasmids encoding short hairpin RNA (*sh*RNA) for mouse *Gpx4* (*shGpx4*: TRCN0000076552), mouse *Lpcat3* (*shLpcat3*: TRCN0000121437), and mouse *Atg3* (*shAtg3*: TRCN0000247442) were obtained from MilliporeSigma. Packaging vector psPAX2 (ID 12260), envelope vector pMD2.G (ID 12259), and scrambled *sh*RNA plasmid (SC: ID 1864) were obtained from Addgene. HEK293T cells in 10 cm dishes were transfected using 50 μL 0.1% polyethylenimine, 200 μL 0.15 M sodium chloride, and 500 μL Opti-MEM (with HEPES, 2.4 g/L sodium bicarbonate, and L-glutamine; Gibco 31985) with 2.66 μg of psPAX2, 0.75 μg of pMD2.G, and 3 μg of either scrambled or *Gpx4*/*Lpcat3*/*Atg3* *sh*RNA plasmids. After 48 hr, growth media were collected and filtered using 0.22 μm vacuum filters to prepare the viral media. Because C2C12 cells are less prone to infection when differentiated, we incubated C2C12 cells in the viral media simultaneous to differentiation (days 1–3). To ensure that only cells infected with *sh*RNA vectors were viable, cells were selected with puromycin throughout differentiation.

## Measurements of myotube diameter

Images of myotubes were visualized at ×20 magnification using an inverted light microscope and captured with a camera (DP74, Olympus). Myotube diameter was measured for at least 100 myotubes in each group (five random fields per well, three wells per experiment, repeated at least three times) using ImageJ software. The average diameter per myotube was calculated as the mean of 10 short-axis measurements taken along the length of the myotube.

## Assessment of cell death

Cell death levels were examined by counting the numbers of cells with trypan blue staining. The cells were trypsinized and stained with 0.2% trypan blue for 5 min. Stained and non-stained cells were counted under a microscope using a hemocytometer.

## Immunofluorescence

C2C12 myotubes were fixed with 4% paraformaldehyde for 10 min and permeabilized with 0.2% Triton X-100 for 15 min. After blocking with bovine serum albumin, immunocytochemistry was performed with anti-HNE (ab48506, Abcam), anti-lysosome-associated membrane protein 2 (Lamp-2) (NB300-591, Novus), and Alexa Fluor-conjugated secondary antibodies Alexa Fluor 568 (ab175472, abcam), Alexa Fluor 488 (ab150073, abcam), and DAPI (D1306, Invitrogen). Images were captured using a 63×1.4 NA oil immersion objective on a Leica SP5 confocal system (Leica). For an experiment, C2C12 myotubes incubated with erastin with or without pretreatment of BaFA1 (SML1661, MilliporeSigma). Soleus muscles were embedded in optimal cutting temperature gel and sectioned at 10 μm with a cryostat (Microtome Plus). The sections underwent blocking for 1 hr with M.O.M. mouse IgG Blocking Reagent (Vector Laboratories, MKB-2213), 1 hr with primary antibodies (BA.D5, SC.71, BF.F3 all at

1:100 from DSHB). Sections were then probed with the following secondary antibodies: Alexa Fluor 647 (1:250; Invitrogen, A21242), Alexa Fluor 488 (1:500; Invitrogen, A21121), and Alexa Fluor 555 (1:500; Invitrogen, A21426). Negative stained fibers were considered to be IIx. Slides were imaged with an automated wide-field light microscope (Nikon Corp) using a 10× objective lens. CSA and fiber-type composition was then quantified utilizing ImageJ software.

## Mitochondrial respiration measurements

Mitochondrial $O_2$ utilization was measured using the Oroboros $O_2K$ Oxygraphs, as previously described (*Eshima et al., 2020*; *Heden et al., 2019*). Isolated mitochondria were added to the oxygraph chambers containing buffer Z. Respiration was measured in response to the following substrate concentrations: 0.5 mM malate, 5 mM pyruvate, 2 mM ADP, 10 mM succinate, and 1.5 µM FCCP.

## Mitochondrial $H_2O_2$ measurements

Mitochondrial $H_2O_2$ production was measured using the Horiba Fluoromax-4, as previously described (*Eshima et al., 2020*; *Heden et al., 2019*). Briefly, skeletal muscle was minced in mitochondria isolation medium (300 mM sucrose, 10 mM HEPES, 1 mM EGTA) and subsequently homogenized using a Teflon glass system. Homogenates were then centrifuged at 800 × *g* for 10 min, after which the supernatant was taken and centrifuged at 12,000 × *g* for 10 min. The resulting pellet was carefully resuspended in mitochondria isolation medium. $JH_2O_2$ was measured in buffer Z (MES potassium salt; 105 mM, KCl 30 mM, $KH_2PO_4$ 10 mM, $MgCl_2$ 5 mM, and BSA 0.5 mg/ml) supplemented with 10 µM Amplex UltraRed (Invitrogen) and 20 U/mL CuZnSOD in the presence of the following substrates: 10 mM succinate, 100 µM 1,3-bis(2-chloroethyl)-1-nitrosourea (BCNU/carmustine), and 1 µM auranofin. The appearance of the fluorescent product was measured with excitation/emission at 565/600 nm.

## Administration of L-carnosine/*N*-acetylcarnosine in vivo

Carnosine was administered as previously described (*Everaert et al., 2013*). Briefly, young (4-month-old) or old (20-month-old) C57BL/6J mice were supplemented with 80 mM carnosine dissolved in drinking water (pH 7.5) for 2 weeks (1 week of pretreatment and 1 week during HU). Bottles were refreshed two times a week (L-carnosine, C9625, MilliporeSigma), and/or everyday (*N*-acetylcarnosine, 18817, Cayman).

## MDA quantification

MDA content was quantified in fresh gastrocnemius muscles using a lipid peroxidation assay kit (ab118970, Abcam) according to the manufacturer's instruction. Rates of appearance of MDA-thiobarbituric acid adduct were quantified colorimetrically at 532 nm using a spectrophotometer.

## Statistical analyses

Data are presented as means ± SEM. Statistical and power analyses were performed using GraphPad Prism 7.03. Independent sample t-tests (two-sided) were used to compare two groups. For multiple comparisons, one- or two-way analysis of variance (ANOVA) were performed followed by appropriate post hoc tests corrected for multiple comparisons. For all tests $p < 0.05$ was considered statistically significant.

## Materials and correspondence

All newly created materials are available for sharing. Correspondence and requests should be addressed to K Funai.

## Acknowledgements

This research is supported by NIH grants DK107397, DK127979, GM144613, AG074535, AG063077, (to KF), AG050781 (to MJD), HL122863, AG057006 (EJA), AG064078 (to QR), HL149870 (SB), HL139451 (to ZSM), DK130555 (to ADP), AG073493 (to JJP), American Heart Association grants 915674 (to PS), 18PRE33960491 (to ARPV), and 19PRE34380991 (to JMJ), Larry H and Gail Miller Family Foundation (to PJF), University of Utah Center on Aging Pilot Grant (to KF), and Uehara Memorial Foundation (to

HE). We would like to thank Diana Lim from the University of Utah Molecular Medicine Program for assistance with figures.

## Additional information

### Competing interests

MacKenzie J Pearson: is affiliated with Sciex. The author has no financial interests to declare. The other authors declare that no competing interests exist.

### Funding

| Funder | Grant reference number | Author |
|---|---|---|
| National Institutes of Health | DK107397 | Katsuhiko Funai |
| National Institutes of Health | DK127979 | Katsuhiko Funai |
| National Institutes of Health | GM144613 | Katsuhiko Funai |
| National Institutes of Health | AG074535 | Katsuhiko Funai |
| National Institutes of Health | AG063077 | Katsuhiko Funai |
| National Institutes of Health | AG050781 | Micah J Drummond |
| National Institutes of Health | HL122863 | Ethan J Anderson |
| National Institutes of Health | AG057006 | Ethan J Anderson |
| National Institutes of Health | AG064078 | Qitao Ran |
| National Institutes of Health | HL149870 | Sihem Boudina |
| National Institutes of Health | HL139451 | Ziad S Mahmassani |
| National Institutes of Health | DK130555 | Alek D Peterlin |
| National Institutes of Health | AG073493 | Jonathan J Petrocelli |
| American Heart Association | 915674 | Piyarat Siripoksup |
| American Heart Association | 18PRE33960491 | Anthony RP Verkerke |
| American Heart Association | 19PRE34380991 | Jordan M Johnson |
| Larry H. & Gail Miller Family Foundation | Predoctoral fellowship | Patrick J Ferrara |
| University of Utah Center on Aging | Pilot grant | Katsuhiko Funai |
| Uehara Memorial Foundation | Postdoctoral fellowship | Hiroaki Eshima |

The funders had no role in study design, data collection and interpretation, or the decision to submit the work for publication.

## Author contributions
Hiroaki Eshima, Conceptualization, Data curation, Formal analysis, Funding acquisition, Investigation, Visualization, Methodology, Writing – original draft, Project administration; Justin L Shahtout, Conceptualization, Data curation, Formal analysis, Validation, Investigation, Visualization, Methodology, Writing – original draft, Project administration, Writing – review and editing; Piyarat Siripoksup, Conceptualization, Data curation, Formal analysis, Funding acquisition, Validation, Investigation, Writing – review and editing; MacKenzie J Pearson, Data curation, Formal analysis, Investigation, Methodology; Ziad S Mahmassani, Data curation, Formal analysis, Validation, Investigation, Methodology; Patrick J Ferrara, Anthony RP Verkerke, Data curation, Formal analysis, Funding acquisition, Validation, Investigation, Methodology; Alexis W Lyons, Data curation, Formal analysis, Validation, Investigation; John Alan Maschek, Data curation, Formal analysis, Validation, Methodology; Alek D Peterlin, Data curation, Formal analysis, Funding acquisition, Validation; Jordan M Johnson, Data curation, Formal analysis, Funding acquisition, Validation, Methodology; Anahy Salcedo, Data curation, Formal analysis; Jonathan J Petrocelli, Data curation, Formal analysis, Methodology; Edwin R Miranda, Formal analysis, Methodology, Writing – review and editing; Ethan J Anderson, Conceptualization, Resources, Funding acquisition, Methodology, Writing – review and editing; Sihem Boudina, Resources, Methodology; Qitao Ran, Resources, Investigation, Methodology, Writing – review and editing; James E Cox, Funding acquisition, Investigation, Methodology, Writing – review and editing; Micah J Drummond, Conceptualization, Data curation, Formal analysis, Supervision, Funding acquisition, Investigation, Methodology, Writing – original draft, Project administration, Writing – review and editing; Katsuhiko Funai, Conceptualization, Resources, Data curation, Formal analysis, Supervision, Funding acquisition, Validation, Investigation, Visualization, Methodology, Writing – original draft, Project administration, Writing – review and editing

## Author ORCIDs
Hiroaki Eshima ⓘ http://orcid.org/0000-0003-4492-5892
Justin L Shahtout ⓘ http://orcid.org/0000-0003-4111-4293
Alek D Peterlin ⓘ http://orcid.org/0000-0002-2837-7446
Katsuhiko Funai ⓘ http://orcid.org/0000-0003-3802-4756

## Ethics
Informed consent and consent to publish was obtained from subjects. All procedures were approved by institutional IRB at the University of Utah and conformed to the Declaration of Helsinki and Title 45, US Code of Federal Regulations, Part 46, "Protection of Human Subjects."
This study was performed in strict accordance with the recommendations in the Guide for the Care and Use of Laboratory Animals of the National Institutes of Health. All of the animals were handled according to approved institutional animal care and use committee (IACUC) protocols (#20-07007) of the University of Utah.

## Decision letter and Author response
Decision letter https://doi.org/10.7554/eLife.85289.sa1
Author response https://doi.org/10.7554/eLife.85289.sa2

# Additional files

## Supplementary files
• MDAR checklist
• Source data 1. A compulation of all uncropped western blots and genotyping gels with labels.

## Data availability
All data generated or analyzed during this study are included in the manuscript.

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
