## [Editor Report]

This paper is of fundamental importance for its description of the role of lipid peroxidation in the loss of muscle mass and contractile function during aging and hindlimb suspension. The evidence is generally solid though is incomplete in some areas. The paper will be of particular interest to those who study the biology of aging-related muscle dysfunction.

---

## [Decision Letter]

**Decision letter after peer review:**

Thank you for submitting your article "Lipid hydroperoxides promote sarcopenia through carbonyl stress" for consideration by *eLife*. Your article has been reviewed by 2 peer reviewers, including Christopher Cardozo as Reviewing Editor and Reviewer #1, and the evaluation has been overseen by Carlos Isales as the Senior Editor.

Essential revisions:

1) Please discuss further the potential limitation of using 20 month old mice, an age when age-related loss of muscle mass may not yet have appeared.

2) Please consider carefully each of the points raised by the reviewers and listed below and provide a response with your resubmission.

*Reviewer #1 (Recommendations for the authors):*

I feel that addressing the following points would improve the paper.

Introduction (line 79) add mention of effects on muscle contractile function which have greater functional impact.

Line 90 revise to be more clear about these lipids. Are these the most abundant oxidized lipids for which a change was observed?

Line 91 consider adding 'species' after oxidized phosphatidylethanolamine.

Line 104 – the data shown are levels of H202 and electron leaks which are different from production of ATP / utilization of fats and carbohydrates as substrates; please consider clarifying the text.

Line 128 – it looks from the figures like the N=2-3 for supplemental Figure S3F and G please clarify the N here.

Line 156 – this is more of a comment outside of the review but please note that MG132 is a peptide aldehyde and not surprisingly for this class of inhibitors blocks action of other peptidases and proteases such as calpains. Irreversible inhibitors such as epoxymicin are quite a bit more specific.

Line 166 – The change in protein content could be explained by changes in synthesis and/or degradation. I am unclear how the experiment informs the reader about the lipidation state of these proteins. I would revise this section carefully to be more conservative.

Line 252 – I would word this less strongly as there are many published mechanisms for unloading-related and aging-related declines in muscle. The data presented here does certainly support a mechanism that raises LOOH but is not sufficient to confirm it is the primary or only one.

Line 308 – there are alternative mechanisms to consider. Impaired excitation-contraction coupling is one well-known deficit of muscle in aging.

Line 357 – why were mice fasted before euthanasia?

Line 391 – the word ‘column’ seems to be missing.

General comments on the Figures

For studies of myotubes, are the data reflective of a single experiment? Were tubes in one well or multiple wells examined? Adding this information would add to the ability of the reader to made an informed assessment of the data.

*Reviewer #2 (Recommendations for the authors):*

– In all experiments, the ages and sex of the animals should be specified. For example, what is the age and sex relative to the human and mouse samples in Fig. 1A-B?

– Soleus and EDL are muscles with a different contractile properties that lead to a different response in disuse models and aging. In this study, the authors mentioned skeletal muscle in general and didn’t specify which muscle they are talking about in the results section. Please amend the manuscript accordingly.

– There is no figure S7I in the supplemental material.– On lines #174-177, the authors mentioned that the loss of muscle ATG3 protects from disuse-induced atrophy and weakness which can be explained by greater myofiber cross-sectional area. Again, what muscle are the authors talking about? Moreover, there is no myofiber diameter data on the EDL muscle.– In the discussion section, line#274, the authors are considering that N-acetylcarnosine as a potential therapeutic to treat muscle atrophy. I would amend the manuscript to state that the N-acetylcarnosine could be considered as a potential therapeutic to treat disuse-induced muscle atrophy because this is model of atrophy primarily used in this manuscript. Other modes of muscle atrophy may or may not be responsive to N-acetylcarnosine.– The connection between LOOH and the autophagy-lysosomal axis should be better presented in the discussion.– In figure S6H, the error bars are in the wrong place.– Figures S11E and S13F are missing the loading controls.– The confocal images in figure 7E are at low resolution – high-resolution images should be used.

---

## [Author Response]

Essential revisions:1) Please discuss further the potential limitation of using 20 month old mice, an age when age-related loss of muscle mass may not yet have appeared.

Thank you. We now provide this discussion in lines 303-305 and 315-319. We also describe below our ongoing study to address this point (response to Reviewer 2).

2) Please consider carefully each of the points raised by the reviewers and listed below and provide a response with your resubmission.

Our point-by-point responses are shown below.

Reviewer #1 (Recommendations for the authors):I feel that addressing the following points would improve the paper.Introduction (line 79) add mention of effects on muscle contractile function which have greater functional impact.

Thank you for this suggestion. Our introduction has been modified to address this point (lines 77-80).

Line 90 revise to be more clear about these lipids. Are these the most abundant oxidized lipids for which a change was observed?

Thank you. Lipids in Figure 1D represent lipid species whose fold-increases with age was the greatest. We apologize for the confusion (lines 91-95).

Line 91 consider adding 'species' after oxidized phosphatidylethanolamine.

Done.

Line 104 – the data shown are levels of H202 and electron leaks which are different from production of ATP / utilization of fats and carbohydrates as substrates; please consider clarifying the text.

Thank you for this suggestion. We agree that this was a confusing way to describe our findings. We rephrased the sentence to “mitochondrial ROS production”.

Line 128 – it looks from the figures like the N=2-3 for supplemental Figure S3F and G please clarify the N here.

We are grateful for helping us identify this omission. It appears that one of the data points in the third bar of Figure 3F was accidentally omitted when copying and pasting on AI. We revised our figure (n=3 for all groups).

Line 156 – this is more of a comment outside of the review but please note that MG132 is a peptide aldehyde and not surprisingly for this class of inhibitors blocks action of other peptidases and proteases such as calpains. Irreversible inhibitors such as epoxymicin are quite a bit more specific.

Thank you for this comment! We were unaware about these effects of MG132 and consider alternatives for future experiments.

Line 166 – The change in protein content could be explained by changes in synthesis and/or degradation. I am unclear how the experiment informs the reader about the lipidation state of these proteins. I would revise this section carefully to be more conservative.

Agreed. We modified the sentence to describe the possibility that syntheses of these proteins may be affected in addition to degradation (lines 170-173).

Line 252 – I would word this less strongly as there are many published mechanisms for unloading-related and aging-related declines in muscle. The data presented here does certainly support a mechanism that raises LOOH but is not sufficient to confirm it is the primary or only one.

Thank you. We rephrased this sentence accordingly (lines 260-262).

Line 308 – there are alternative mechanisms to consider. Impaired excitation-contraction coupling is one well-known deficit of muscle in aging.

Thank you for this suggestion. We now include a discussion on excitation-contraction coupling as a potential mediator (lines 328-331).

Line 357 – why were mice fasted before euthanasia?

Because all mice have free access to food, any given mice could vary largely with the timing of their last food consumption. We routinely remove food prior to terminal experiments so that no mice had eaten immediately prior to tissue collection (lines 381-384).

Line 391 – the word ‘column’ seems to be missing.

Thank you for this correction (lines 415-417).

General comments on the FiguresFor studies of myotubes, are the data reflective of a single experiment? Were tubes in one well or multiple wells examined? Adding this information would add to the ability of the reader to made an informed assessment of the data.

We apologize for omitting this information. We added more details in the method (lines 456-458).

Reviewer #2 (Recommendations for the authors):– In all experiments, the ages and sex of the animals should be specified. For example, what is the age and sex relative to the human and mouse samples in Fig. 1A-B?

Thank you. We now provide this information in the figure legends.

– Soleus and EDL are muscles with a different contractile properties that lead to a different response in disuse models and aging. In this study, the authors mentioned skeletal muscle in general and didn’t specify which muscle they are talking about in the results section. Please amend the manuscript accordingly.

This is an excellent point. We primarily focused on soleus and EDL muscles (data on EDL muscles are primarily in the supplement as soleus had a more robust change in mass with HU) because they could be used to quantify ex vivo contractile activity, but in no way these two muscles necessarily represent all muscle types throughout the body. We modified our manuscript accordingly.

– There is no figure S7I in the supplemental material.

Thank you. This was our mistake and the text is referring to S7H instead (line 180).

– On lines #174-177, the authors mentioned that the loss of muscle ATG3 protects from disuse-induced atrophy and weakness which can be explained by greater myofiber cross-sectional area. Again, what muscle are the authors talking about? Moreover, there is no myofiber diameter data on the EDL muscle.

Thank you. We modified the text accordingly that our findings for muscle mass was in soleus and force-generating capacity in both soleus and EDL muscles (lines 179-183). We chose to section soleus only because masses of EDL was not significantly reduced by HU. For preceding figures (GPx4+/- and GPx4-MKO) where deletion of GPx4 made atrophy worse in both soleus and EDL muscles we performed the fiber-type/CSA analyses in both muscles. For ATG3-MKO, GPx4Tg, L-carnosine and N-acetylcarnosine studies where these interventions prevented the atrophy, which was only observed in soleus (largely because masses of EDL was not significantly reduced by HU), we focused our fiber-type/CSA analyses in soleus.

– In the discussion section, line#274, the authors are considering that N-acetylcarnosine as a potential therapeutic to treat muscle atrophy. I would amend the manuscript to state that the N-acetylcarnosine could be considered as a potential therapeutic to treat disuse-induced muscle atrophy because this is model of atrophy primarily used in this manuscript. Other modes of muscle atrophy may or may not be responsive to N-acetylcarnosine.

Thank you. We amended this phrase (lines 281-283).

– The connection between LOOH and the autophagy-lysosomal axis should be better presented in the discussion.

Thank you. We now add these discussions in lines 280-291.

– In figure S6H, the error bars are in the wrong place.

Corrected.

– Figures S11E and S13F are missing the loading controls.

Loading controls have now been added.

– The confocal images in figure 7E are at low resolution – high-resolution images should be used.

We utilized a confocal microscope at our central core facility. We currently do not have immediate access to a microscope with better resolution, but when we will perform further studies on LOOH and lysosome we will seek to obtain better images.